# The Sexual Double Standard toward Non-Heterosexual Populations: Evaluations of Sexually Active Gay Men and Lesbian Women

**DOI:** 10.3390/bs14080706

**Published:** 2024-08-13

**Authors:** Michael Marks, Serina Padgett

**Affiliations:** Department of Psychology, New Mexico State University, Las Cruces, NM 88003, USA; spadgett@nmsu.edu

**Keywords:** sexual double standard, gender, sex, sexuality, gender norms, masculinity, femininity

## Abstract

We examined the sexual double standard (SDS) toward sexually active gay men and lesbian women and the role that participants’ masculinity and femininity played in their evaluations. We hypothesized that there would be a reverse SDS in which highly sexually active lesbian women would be evaluated positively and highly sexually active gay men negatively, with both being evaluated more negatively than less sexually active gay men and lesbians. We also hypothesized that masculinity would moderate this effect, with participant masculinity being positively related to stronger negative evaluations of highly sexually active gay targets and more positive evaluations of highly sexually active lesbian targets. Results indicate a weak SDS in the areas of likability and morality, with highly sexually active lesbians being liked by masculine participants the most and highly sexually active gay men being rated as the least moral. The SDS appears to be influenced by expectations of gender roles and may be reversed for gay men and lesbian women because of these expectations.

## 1. Introduction

The sexual double standard (SDS) is the notion that men and women are evaluated differently with regard to their level of sexual activity [1]. According to the SDS, highly sexually active men are evaluated positively and highly sexually active women are evaluated negatively [2,3,4]. To date, most research on the SDS has examined the perceptions of heterosexual populations; there is little research on how the SDS may be applied toward non-heterosexual populations. This is a particularly interesting and important consideration, as a better understanding of what influences the SDS may help us to understand its mechanics and how it impacts evaluations of a broader range of people. The goal of the present research is to examine whether a SDS is applied to sexually active gay men and lesbian women, an understudied population in SDS research. We also consider the role of the participants’ masculinity and femininity in their evaluations of sexually active gay men and lesbians. We begin with a review of SDS research.

The SDS has been a now-you-see-it-now-you-do-not phenomenon in contemporary research. Some research has demonstrated that people evaluate men and women based on their sexual activity but do not necessarily hold them to different standards [1]. Other research has demonstrated a weak SDS, whereby both men and women are derogated for having many partners but to different degrees [5]. In other words, both men and women were derogated for having many sexual partners, but sexually active women were derogated to a greater degree than were sexually active men. This represents a change from historical patterns of the SDS in which sexually active men are rewarded and sexually active women are derogated [6,7]. With respect to this SDS pattern, some scholars have suggested the existence of, but do not necessarily document, a more traditional (strong) SDS, whereby sexually active men are rewarded and sexually active women derogated [8].

Although there are many reasons for these discrepancies, culture may be one factor that dictates whether a SDS emerges. In Western-European societies, men are socialized to be dominant, independent, and to avoid displays of affection for other men, whereas women are socialized to be sensitive, passive, and chaste [9,10,11]. Masculine norms also encourage enhancing one’s sexual reputation through seduction or exhibiting sexual prowess [12,13]. With respect to sexual behaviors, societies construct hierarchies of sexual value, where a small subset of sexual practices and identities (termed the “charmed circle”) are deemed superior to others [14]. These hierarchies, which reflect broader societal norms and moral judgments, limit acceptable forms of sex to that which is heterosexual, monogamous, married, procreative, and vanilla. All other forms of sex are viewed as deviant and those who enact them (e.g., gay or paid sex) are subject to various forms of societal stigmatization [14]. Notably, however, although non-heterosexual sex is stigmatized, lesbians are viewed more favorably than gays [15,16] because lesbian sex is non penetrative and is less associated with sexually transmitted diseases than gay sex [17].

Cultural manifestation of the SDS may further be impacted by adherence to gender roles, specifically with respect to heteronormativity. A high level of sexual activity in heterosexual women violates gender roles, but a high level of sexual activity is gender normative for heterosexual men; this may partially explain the SDS. If nonconformity to social norms plays a major factor in the evaluation of others, then it is possible that sexually active gay men and lesbian women will be evaluated in ways that belie a traditional SDS, as: (a) non-heterosexuality does not conform to traditional gender norms [18,19], and (b) people are generally less accepting of those who violate gender roles [9,20,21].

Reactions toward gender-nonconforming individuals and non-heterosexual individuals can be negative. However, lesbian women experience less verbal harassment and threats than gay men [22,23]. In general, there is greater acceptance of gender nonconformity in women than in men [24], possibly exacerbating the already lesser acceptability of gay men compared to lesbians [15,16]. A stigma is associated with gender nonconformity, as is an overall negative response to gender-nonconforming individuals [18,20].

In summary, gender role conformity plays an important role in people’s evaluations of others [18,20]. Gay men and lesbian women are less likely to conform to typical perceptions of gender roles and thus may be evaluated negatively (compared to heterosexuals) by others. Since the SDS may be an extension of this idea, we examined how sexually active gay men and lesbian women are evaluated by others. Sexually active lesbians deviate from the female gender role of chastity and sexual passivity, whereas sexually active gay men both deviate from gender roles (via their sexual orientation) and follow them (with respect to their level of sexual activity [10]). However, these gender roles do not necessarily translate to non-heteronormative populations, as described earlier (i.e., although lesbians perhaps violate a greater number of gender norms than gay men, they are not nearly as stigmatized for them as are gay men).

Another aspect of assessment we examined is whether the evaluator’s levels of masculinity and femininity affect the way they view others. Masculinity and femininity are socially constructed norms that convey the types of behaviors commonly associated with men and women, respectively. There are several factors that affect social perceptions of sex, including gender, race, culture, and class [25]. These factors shape how we perceive masculinity, femininity, and each other, as well as what behaviors and actions we categorize as gender-appropriate. For example, men are expected to maintain not only the physical, but also the emotional, hegemonic characteristics of masculinity, otherwise they may be perceived as emasculated. Masculinity is associated with having many lifetime sex partners and the positive aspects of sex in relation to femininity [26,27]. As such, people high in masculinity hold more positive attitudes toward sex [12,13].

We hypothesize a reverse SDS in which the highly sexually active lesbian target will be evaluated positively and the highly sexually active gay target negatively, with both being evaluated more negatively than less sexually active gay and lesbian targets (H1). We also hypothesized that masculinity would moderate this effect, with participant masculinity being positively related to stronger negative evaluations of the highly sexually active gay target and more positive evaluations being made of the highly sexually active lesbian target (H2). Finally, we hypothesize that the gay male target will be evaluated most negatively of all targets given that gay men are generally evaluated more harshly than lesbian women [28] (H3).

## 2. Materials and Methods

### 2.1. Design

The independent variables for this experiment are sexual activity (highly sexually active vs. moderately sexually active) and gender (male vs. female) of a hypothetical sexually active person. Operationally defined, “high” sexual activity is having 19 sexual partners and “moderate” sexual activity is having 7 sexual partners (numbers adapted from [1]). Participant masculinity and femininity were also predictors. The dependent variables were the means of the likeability, morality, desirability (as a partner/friend), success, and intelligence items. Data were analyzed using a series of hierarchical linear regressions.

### 2.2. Participants

An a priori power analysis using G*Power software version 3.1.9.2 suggested that 132 participants would be necessary to detect large effects (F^2^ = 0.15), with 90% power at an alpha cutoff of 0.05 in a regression with 9 tested predictors (4 main effects and 5 interactions). Anticipating some attrition, we initially recruited 172 students from a large southwestern university who participated for partial credit in their introductory psychology class. After excluding 7 participants for failing to complete the survey and 8 for failing to follow directions, our final sample contained 157 participants (resulting in a power of 0.93, which we did not deem excessive relative to our a priori analysis). The average participant age was 20.1 years (*SD* = 5.03, range 17–46), and 67% were women. Approximately 40% of the participants were European American, 48% Hispanic American, 2% African American, and 10% of other ethnicities or who did not specify their ethnicity. Finally, 91% of participants were heterosexual, and 53% were in a romantic relationship.

### 2.3. Materials

#### 2.3.1. Personal Attributes Questionnaire (PAQ) [29]

The PAQ is designed to measure an individual’s levels of masculinity and femininity and contains 24 pairs of characteristics on which the respondent is asked to rate themselves on a 5-point scale. Each pair of characteristics is designed so that the respondent cannot be both at the same time (e.g., not at all aggressive, A…B…C…D…E, very aggressive).

#### 2.3.2. Sexual Double Standard

To measure the SDS, four vignettes (see Appendix A) were created to assess the respondents’ attitudes toward hypothetical sexually active gay men and lesbian women. Information about the individual’s sexuality and recent sexual activity was given amid information about their profession, hobbies, and personality. The vignette was followed by 15 questions designed to assess the target in five areas (with internal consistencies for the current sample): likeability (α = 0.77), morality (α = 0.77), desirability as a partner or friend (α = 0.81), success (α = 0.79), and intelligence (α = 0.76; see Appendix B for the items within each domain). These items, adapted from [1], demonstrated sensitivity to variation in both gender and the number of sexual partners, which are variables central to the SDS.

### 2.4. Procedure

This study was listed using SONA Systems software as a project on person perception. Upon arriving to the lab, participants received an informed consent form that they read and signed prior to beginning the experiment. Participants then completed questionnaire packets containing a demographics form, the PAQ, and one of four randomly assigned hypothetical target vignettes accompanied by the evaluative questions.

## 3. Results

### 3.1. Analysis Strategy

For each of the five outcome variables, a three-step hierarchical linear regression was conducted. Step 1 included participant gender and age as controls, the target sex (man/woman), and the target number of partners (high/typical). Step 2 included two-way interaction terms for the target sex by partners, the target sex by the each of the PAQ masculinity and femininity scores, between target partners, and the PAQ scores (five total interactions). Step 3 included the three-way interaction terms for each of the PAQ scores by the target sex by partners (two total interactions).

### 3.2. Likability

The main effect and control variables entered in Step 1 explained 5.2% of the variance in likability ratings (*F* (4, 152) = 2.06, Δ*R*^2^ = 0.052, *p*_change_ = 0.088). Women rated the target person as more likeable than did men (*β* = 0.168 and *p* = 0.036). Target persons with more partners were rated as marginally less likable than those with fewer partners (*β* = −0.132 and *p* = 0.099). In Step 2, adding the two-way interaction terms resulted in an additional 1.9% of the variance in likability ratings being explained (*F* (5, 147) = 0.60, Δ*R*^2^ = 0.019, and *p*_change_ = 0.701). No two-way interactions terms were significant. In Step 3, adding the three-way interaction terms resulted in an additional 2.7% of the variance in likability ratings being explained (*F* (2, 145) = 2.15, Δ*R*^2^ = 0.027, and *p*_change_ = 0.120) (see Table 1). There was a significant three-way interaction between the target sex, target partners, and participant masculinity (*β* = 0.252 and *p* = 0.040), whereby participants higher in masculinity rated lesbian women as more likable as their number of partners increased (*b_slope_* = 0.730 and *p_slope_* = 0.004) relative to the other three groups, for which there was no relationship with participant masculinity (see Figure 1).

### 3.3. Morality

The main effect and control variables entered in Step 1 explained 9.8% of the variance in the morality ratings (*F* (4, 152) = 4.11, Δ*R*^2^ = 0.098, and *p*_change_ = 0.003). Women rated the target person as more moral than did men (*β* = 0.252 and *p* = 0.001). Target persons with more partners were rated as marginally less moral than those with fewer partners (*β* = −0.152 and *p* = 0.052). In Step 2, adding the two-way interaction terms resulted in an additional 2.4% of the variance in morality ratings being explained (*F* (5, 147) = 0.79, Δ*R*^2^ = 0.024, and *p*_change_ = 0.560). There was a marginal interaction between the target sex and partners (*β* = −0.231 and *p* = 0.100), whereby lesbian women were rated as less moral as the number of partners increased (*b_slope_* = −0.373 and *p_slope_* = 0.051), but gay men even more so (*b_slope_* = −1.016 and *p_slope_* < 0.001), evidencing a reverse SDS (see Figure 2). In Step 3, adding the three-way interaction terms resulted in an additional 0.1% of the variance in morality ratings being explained **F* (2, 145) = 0.60, Δ*R*^2^ = 0.009, and *p*_change_ = 0.499). No three-way interaction terms were significant (see Table 2).

### 3.4. Partner/Friend Desirability

The main effect and control variables entered in Step 1 explained 9.6% of the variance in morality ratings (*F* (4, 152) = 4.04, Δ*R*^2^ = 0.096, and *p*_change_ = 0.004). Women rated the target person as more desirable than did men (Δ*R*^2^ = 0.169 and *p* = 0.031). Target persons with more partners were rated as less desirable than those with fewer partners (Δ*R*^2^ = −0.240 and *p* = 0.002). In Step 2, adding the two-way interaction terms resulted in an additional 3.2% of the variance in morality ratings being explained (*F* (5, 147) = 1.10, Δ*R*^2^ = 0.032, and *p*_change_ = 0.366). No two-way interactions terms were significant. In Step 3, adding the three-way interaction terms resulted in an additional 1.2% of the variance in morality ratings being explained (*F* (2, 145) = 0.60, Δ*R*^2^ = 0.012, and *p*_change_ = 0.362). No three-way interaction terms were significant (see Table 3).

### 3.5. Success

The main effect and control variables entered in Step 1 explained 7.5% of the variance in success ratings (*F* (4, 152) = 3.08, Δ*R*^2^ = 0.075, and *p*_change_ = 0.018). Women rated the target person as marginally more successful than did the men (Δ*R*^2^ = 0.151 and *p* = 0.057). Target persons with more partners were rated as marginally less successful than those with fewer partners (*β* = −0.138 and *p* = 0.082). In Step 2, adding the two-way interaction terms resulted in an additional 4% of the variance in success ratings being explained (*F* (5, 147) = 1.32, Δ*R*^2^ = 0.040, and *p*_change_ = 0.258). No two-way interaction terms were significant. In Step 3, adding the three-way interaction terms resulted in an additional 0.06% of the variance in morality ratings being explained (*F* (2, 145) = 0.50, Δ*R*^2^ = 0.006, and *p*_change_ = 0.608). No three-way interaction terms were significant (see Table 4).

### 3.6. Intelligence

The main effect and control variables entered in Step 1 explained 7.9% of the variance in intelligence ratings (*F* (4, 152) = 3.25, Δ*R*^2^ = 0.079, and *p*_change_ = 0.014). Female targets were rated as marginally more intelligent than male targets (Δ*R*^2^ = 0.133 and *p* = 0.091). Target persons with more partners were rated as less intelligent than those with fewer partners (*β* = −0.207 and *p* = 0.009). In Step 2, adding the two-way interaction terms resulted in an additional 1.5% of the variance in intelligence ratings being explained (*F* (5, 147) = 0.49, Δ*R*^2^ = 0.015, and *p*_change_ = 0.782). No two-way interaction terms were significant. In Step 3, adding the three-way interaction terms resulted in an additional 0.07% of the variance in morality ratings being explained (*F* (2, 145) = 0.57, Δ*R*^2^ = 0.007, and *p*_change_ = 0.568). No three-way interaction terms were significant (see Table 5).

### 3.7. Summary

To summarize the results by hypothesis, Hypothesis 1 was supported only in the domain of Morality, where lesbian women were rated less negatively than gay men in the high sexual partner condition, evidencing a reverse SDS. Hypothesis 2 was supported only in the domain of Likability, whereby participants higher in masculinity rated lesbian women as more likable as their number of partners increased relative to the other three groups, for whom there was no relationship with participant masculinity. In other words, there was a stronger preference for the highly sexually active lesbian target relative to other conditions by more masculine participants. Finally, Hypothesis 3 was only weakly supported in the domain of Intelligence, with lesbian women rated as marginally more intelligent than gay men. There were no Hypothesis supporting or refuting results in the domains of Desirability and Success.

## 4. Discussion

We examined whether the SDS is applied to gay men and lesbian women, hypothesizing that there would be an overall weak SDS. In a weak SDS, both sexually active men and women are rated differently than one another but not in opposite directions. We predicted that gay men would be evaluated more negatively than lesbian women and that the participants’ levels of masculinity and femininity would moderate their evaluations of the sexually active target. Results indicated the presence of a weak double standard in two domains: In the domain of Likability, participants higher in masculinity rated lesbian women as more likable as their number of partners increased relative to other conditions. Masculinity was not related to the ratings in the other three groups. In the domain of Morality, both gay men and lesbian women who had more partners were rated as less moral but this effect was larger for gay men than lesbian women.

The results partially support the existence of the weak double standard described by Marks and Fraley [1], in which sexually active men and women are both derogated but one gender is derogated more so than the other. Our findings in the two domains exhibit a similar theme of differences being more in degree of valence rather than valence per se. These findings contradict the traditional double standard theory, indicating that there may be a reversed double standard toward non-heterosexual populations, at least in some evaluative domains. In the domain of Morality, the gay male target with a high number of partners invoked the stereotype of the stigmatized “Gay promiscuous man,” eliciting negative evaluations of immorality by the participants [30]. Although heterosexual, highly sexually active men often receive negative evaluations in SDS research (e.g., [1]), where the added social stigma of being gay may exacerbate those negative evaluations. While stereotypes of lesbian women also invoke promiscuity [30], the eroticization of lesbian women [31] possibly attenuates negative evaluations.

In the other example of a weak SDS in the current research, the participants’ levels of masculinity moderated the relationship between the target sex, partners, and ratings in the domain of Likability. More masculine participants rated active lesbian women as more likable as their sexual activity increased, whereas evaluations of other groups were not affected by the predictors. These findings indicate masculinity may partially contribute to the evaluations of others in terms of the SDS. A correlate of masculinity is pride in manhood [32]. Since sexual agency is a quality associated with manhood [33], this, coupled with the eroticization of lesbian women [31], could have led masculine participants—regardless of gender—to see the highly sexually active lesbian woman as having several valued qualities. Further, since sexual agency and sexuality are qualities that many masculine people endorse [34], it is possible that the lesbian woman target person (i.e., erotic and sexual) was the target that masculine participants saw as most similar to themselves, and similarity has long been recognized as one of the strongest predictors of likability [35].

Why a weak SDS? Research on the SDS has examined the function of expectations and roles as factors of the double standard. Baumeister and Twenge [3] reported that the SDS may be caused by women who suppress their sexuality as a means of negotiating with men. Marks and Fraley [36] found that participants were more likely to notice and remember instances that confirmed the double standard and ignored or did not notice situations that disconfirmed it. Since the SDS is influenced by situation [7], expectations may play an important role in the evaluation of others. If an individual is primed to notice gender roles or expect a double standard, they may be more likely to notice deviations in gender roles; hence, a weak, rather than strong, double standard.

Further, expectations for an individual to fulfill certain gender roles may cause a feeling of unacceptance. Although this affects heterosexual individuals [2,11], gay men and lesbian women are compounded by multiple stereotypes and behavioral expectations from others. Non-heterosexuality is marked by gender role deviation [10,15,17]. This deviation leads to unfavorable views and opinions of gay men and lesbian women by others. Gay men and lesbian women often have low self-esteem and mental health challenges due to public scrutiny [37,38]. Our findings reflect the trend of negative evaluations for most highly sexually active individuals [1,5,39,40].

### Limitations and Future Directions

Some limitations of this research deal with generalizability. The participants were mostly college students in their early twenties, who are generally more open-minded than older adults who grew up in a different era when non-heterosexuality may have been less accepted. As a result, these findings may be limited to the college student population. Furthermore, gender roles may have changed from the time when the Personal Attributes Questionnaire was created in the 1970s. Men and women adhered to stricter gender roles then, whereas today some of the same characteristics are common in both genders, such as having leadership ability, being active, and being self-confident. To assuage this concern, a review of the literature suggests that the PAQ is still one of the most frequently used assessments of gender with its main dimensions of masculinity and femininity [41], and it still shows convergent validity with other self-report measures of gender self-concept [42].

We would also like to note some caveats with respect to the current methods and results. First, the current study did not have enough statistical power to assess four-way interactions between participant gender, masculinity/femininity, target sex, and target partners. This would be an interesting analysis in that one could assess the impact of whether participants’ gender ideology aligns with their biological sex. Related to this caveat, two thirds of our sample consisted of women, which could have resulted in decreased generalizability of our findings to more general populations. Third, we described the targets as having a business degree, which, in hindsight, may be associated more with masculinity than femininity according to gender stereotypical perspectives. Finally, we also would like to note that our current results featured few significant effects, so the current results should be treated with caution until further research is conducted and there is a larger body of SDS research on non-heterosexual populations.

Future research could examine current gender roles and reevaluate the SDS with relation to these gender roles. Another possible study could examine the SDS toward gay men and lesbian women through evaluation by a non-heteronormative sample. Previous research has examined the SDS in heterosexual populations where heterosexual participants conducted the evaluations. People evaluating members of their ingroup group may influence results; the current research was focused on members of one group being evaluated by members of a different group. To briefly explore this idea, we reanalyzed the data using only the heterosexual participants^1^. There were not enough non-heterosexual participants in the sample for a full analysis, so we opted to run the analyses without them to explore if there were any major differences in significant effects. However, there were not. It would be interesting to examine whether the SDS manifests similarly in a full sample of non-heterosexual respondents when they are evaluating members of a non-heterosexual population.

In closing, we found a weak SDS only in two of five evaluative domains, suggesting the SDS may not be as prevalent in evaluations of non-heteronormative populations. Studying the SDS toward (or in) non-heteronormative samples is still an understudied area, and many more studies are needed before we can even begin to understand how the SDS is generated, sustained, and impacts non-heterosexual individuals. We hope the current research inspires and generates further research, studies, and discussion in and about this topic.

## Figures and Tables

**Figure 1 behavsci-14-00706-f001:**
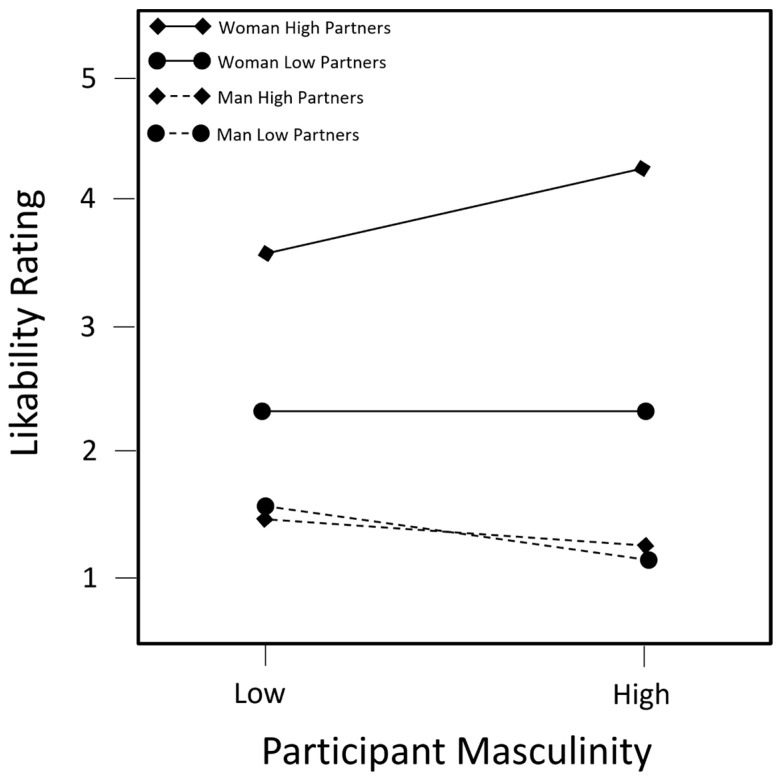
Likability ratings of target persons based on participant masculinity, target sex, and number of partners. Note: *N* = 157.

**Figure 2 behavsci-14-00706-f002:**
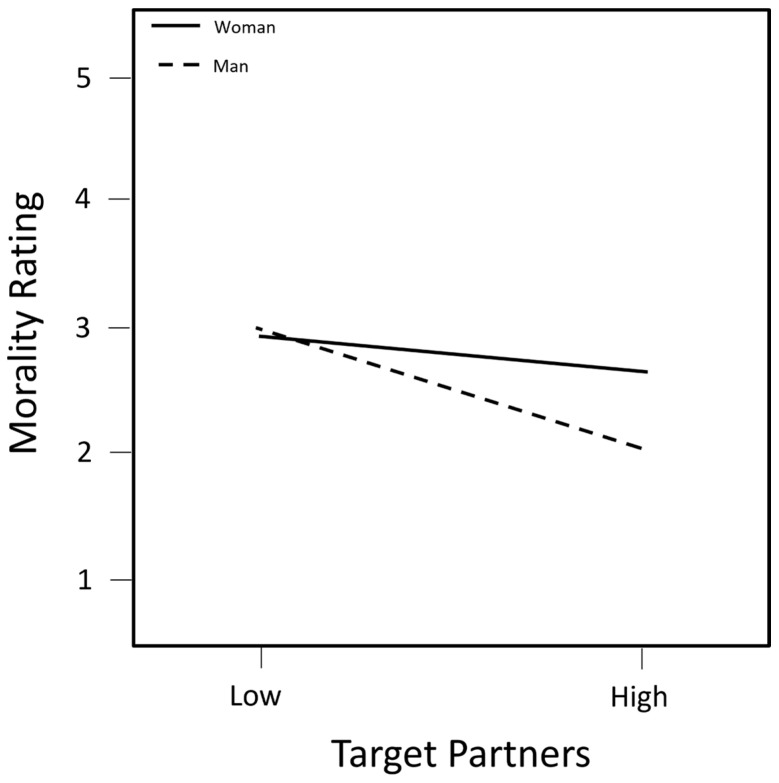
Morality ratings of the target persons based on the target sex and number of partners. Note: *N* = 157.

**Table 1 behavsci-14-00706-t001:** Regression analysis of the likability ratings, as predicted by the control and manipulated variables.

Predictor	*B*	*β*	*p*	95% CI *B*	Δ*R*^2^
Step 1					0.05
Participant Gender	0.492	0.168	0.036	[0.032, 0.952]	
Participant Age	0.008	0.031	0.700	[−0.035, 0.052]	
Target Sex	−0.029	−0.011	0.894	[−0.462, 0.404]	
Target Partners	−0.364	−0.132	0.099	[−0.798, 0.070]	
Step 2					0.02
Target Sex X Partners	0.026	0.008	0.955	[−0.872, 0.923]	
Target Sex X Femininity	0.192	0.094	0.274	[−0.154, 0.538]	
Target Sex X Masculinity	0.097	0.046	0.601	[−0.267, 0.461]	
Target Partners X Fem.	0.071	0.036	0.680	[−0.267, 0.408]	
Target Partners X Masc.	−0.162	−0.092	0.303	[−0.472, 0.148]	
Step 3					0.03
Target Sex X Part. X Fem.	0.004	0.399	0.993	[−0.785, 0.792]	
Target Sex X Part. X Masc.	0.795	0.252	0.040	[0.037, 1.55]	
Total *R*^2^					0.10

Note: *N* = 157 and CI = confidence interval.

**Table 2 behavsci-14-00706-t002:** Regression analysis of the morality ratings, as predicted by the control and manipulated variables.

Predictor	*B*	*β*	*p*	95% CI *B*	Δ*R*^2^
Step 1					0.10
Participant Gender	0.654	0.252	0.001	[0.256, 1.05]	
Participant Age	−0.013	−0.054	0.485	[−0.051, 0.024]	
Target Sex	0.053	0.022	0.779	[−0.322, 0.428]	
Target Partners	−0.373	−0.152	0.052	[−0.749, 0.003]	
Step 2					0.03
Target Sex X Partners	−0.643	−0.231	0.100	[−1.42, 0.131]	
Target Sex X Femininity	0.014	0.008	0.926	[−0.285, 0.313]	
Target Sex X Masculinity	0.037	0.019	0.0.818	[−0.278, 0.351]	
Target Partners X Fem.	0.111	0.063	0.750	[−0.278, 0.351]	
Target Partners X Masc.	−0.140	−0.089	0.302	[−0.408, 0.128]	
Step 3					0.01
Target Sex X Part. X Fem.	−0.241	−0.077	0.489	[−0.929, 0.446]	
Target Sex X Part. X Masc.	0.323	0.115	0.336	[−0.338, 0.983]	
Total *R*^2^					0.13

Note: *N* = 157 and CI = confidence interval.

**Table 3 behavsci-14-00706-t003:** Regression analysis of the desirability ratings, as predicted by the control and manipulated variables.

Predictor	*B*	*β*	*p*	95% CI *B*	Δ*R*^2^
Step 1					0.10
Participant Gender	0.566	0.169	0.031	[0.052, 1.08]	
Participant Age	−0.002	−0.005	0.948	[−0.050, 0.047]	
Target Sex	0.110	0.035	0.655	[−0.374, 0.593]	
Target Partners	−0.755	−0.240	0.002	[−1.24, −0.271]	
Step 2					0.03
Target Sex X Partners	−0.152	−0.042	0.762	[−1.14, 0.842]	
Target Sex X Femininity	0.228	0.097	0.241	[−0.155, 0.612]	
Target Sex X Masculinity	−0.049	−0.020	0.811	[−0.452, 0.354]	
Target Partners X Fem.	0.210	0.092	0.270	[−0.164, 0.583]	
Target Partners X Masc.	−0.197	−0.097	0.260	[−0.540, 0.147]	
Step 3					0.01
Target Sex X Part. X Fem.	−0.608	−0.151	0.174	[−1.49, 0.272]	
Target Sex X Part. X Masc.	0.186	0.052	0.664	[−0.660, 1.03]	
Total *R*^2^					0.14

Note: *N* = 157 and CI = confidence interval.

**Table 4 behavsci-14-00706-t004:** Regression analysis of the success ratings, as predicted by the control and manipulated variables.

Predictor	*B*	*β*	*p*	95% CI *B*	Δ*R*^2^
Step 1					0.08
Participant Gender	0.386	0.151	0.057	[−0.012, 0.784]	
Participant Age	−0.019	−0.079	0.316	[−0.056, 0.018]	
Target Sex	0.369	0.153	0.053	[−0.005, 0.743]	
Target Partners	−0.332	−0.138	0.082	[−0.707, 0.043]	
Step 2					0.04
Target Sex X Partners	0.024	0.009	0.951	[−0.743, 0.790]	
Target Sex X Femininity	0.209	0.117	0.165	[−0.087, 0.504]	
Target Sex X Masculinity	0.114	0.061	0.471	[−0.197, 0.425]	
Target Partners X Fem.	0.138	0.079	0.346	[−0.150, 0.426]	
Target Partners X Masc.	−0.211	−0.136	0.118	[−0.476, 0.054]	
Step 3					0.01
Target Sex X Part. X Fem.	−0.234	−0.076	0.497	[−0.915, 0.447]	
Target Sex X Part. X Masc.	0.245	0.089	0.461	[−0.410, 0.899]	
Total *R*^2^					0.12

Note: *N* = 157 and CI = confidence interval.

**Table 5 behavsci-14-00706-t005:** Regression analysis of the intelligence ratings, as predicted by the control and manipulated variables.

Predictor	*B*	*β*	*p*	95% CI *B*	Δ*R*^2^
Step 1					0.08
Participant Gender	0.232	0.107	0.173	[−0.103, 0.568]	
Participant Age	0.000	−0.001	0.990	[−0.023, 0.031]	
Target Sex	0.272	0.133	0.091	[−0.044, 0.588]	
Target Partners	−0.422	−0.207	0.009	[−0.739, −0.106]	
Step 2					0.02
Target Sex X Partners	0.011	0.005	0.973	[−0.645, 0.667]	
Target Sex X Femininity	0.091	0.060	0.478	[−0.162, 0.344]	
Target Sex X Masculinity	0.068	0.044	0.612	[−0.198, 0.334]	
Target Partners X Fem.	0.029	0.020	0.817	[−0.218, 0.275]	
Target Partners X Masc.	−0.149	−0.113	0.197	[−0.375, 0.078]	
Step 3					0.01
Target Sex X Part. X Fem.	−0.314	−0.120	0.288	[−0.896, 0.268]	
Target Sex X Part. X Masc.	0.013	0.006	0.963	[−0.547, 0.573]	
Total *R*^2^					0.10

Note: *N* = 157 and CI = confidence interval.

## Data Availability

Data and materials are available from the first author upon request.

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
