# Peer review of "The Sexual Double Standard toward Non-Heterosexual Populations: Evaluations of Sexually Active Gay Men and Lesbian Women"

_behavsci, 2024, doi:10.3390/bs14080706_

Round 1

Reviewer 1 Report

Comments and Suggestions for Authors

I enjoyed reading this paper, which presents an interesting idea about testing whether there is a sexual double standard for gay men and lesbians in relation to number of sexual partners. But I’m not completely sure that their predictions follow from what they found in the literature.

52-57: “Similar research on men suggests that those who adhere to traditional gender roles are less satisfied with their romantic relationships than those who adopt unconventional gender roles. Women, however, are more satisfied when their partners adhere to traditional masculine roles. This suggests that although an individual may not be satisfied with adhering to traditional gender roles, their partners prefer that they do.”

I don’t see how the final sentence follows, and I’m also confused about the second sentence.

The first sentence says that masculine/traditional men less satisfied in relationship than unmasculine/feminine/untraditional men; it says nothing about what their partners prefer. The second sentence says that women like more masculine men (or feminine women?). The third sentence says that these ideas suggest that individuals prefer that their partner be traditionally masculine if a man (or traditionally feminine if a woman?—is that found in the research?).  But this seems only true for women, from what was written. Also, it is only masculine/traditional men who are not satisfied with their romantic relationships; an earlier sentence (lines 49-52) says that traditional feminine women are less satisfied in their lives (and says nothing about their relationships). The linkages between ideas are, thus, a bit messy here.

58-67: The idea here is that, because gay men and lesbians, in being gay and lesbian, go against gender and (for lesbians with many sex partners) sexual norms, the sexual double standard may apply differently to them than to heterosexuals, for whom men having lots of sex is fine, women doing so not fine.

We are told that lesbians receive less verbal harassment than gay men, and are more accepted, so should be less negatively affected by sexual double standard than are gay men.

The authors state that “Gay men and lesbian women are less likely to conform to traditional gender roles [compared to heterosexuals], and thus may be evaluated negatively (compared to heterosexuals) by others” (lines 75-76). But we are not told if John (the gay man) or Joanna (the lesbian) is feminine or masculine, so participants have to rely on stereotype to interpret John and Joanna as gender nonconforming.

Some issues:

(1) The authors’ assumption that sexual orientation and gender nonconformity are perceived the same seems belied in research when these are separated:  “Most surprisingly, for boys, the straight individual who was non-conforming in appearance was rated less acceptable than either the gay individual who conformed to gender norms or was gender non-conforming in choice of activity.” (Horn, 2007, p. 363).

(2) Bell, Weinberg & Hammersmith (1982) found that most gay men and lesbians were gender conforming (did not stand out as gender nonconforming); I think about 23% of their male participants (not randomly selected) were gender nonconforming, but don’t quote me. I think the original Kinsey studies also found that most MSM and WSW were normative gender-wise.

All this means that the authors are working on the idea that their participants have a gender-nonconforming stereotype of gay men and lesbians, but they may not. The authors’ questionnaire pushes this stereotype more for the lesbian than for the gay man, as a business degree seems more “masculine”. This might be discussed as a limitation.

(3) Lesbians violate gender norms (by not being chaste, not being sexually passive AND by being lesbian AND by being masculine?), and by having sex with more people violate sexual roles (women are not supposed to want sex).

Gay men violate gender norms (by being feminine (?) and by being gay), but having sex with more people supports sexual roles (men are supposed to have lots of sex).

From all this I would expect that lesbians should be MORE negatively evaluated the more sexual partners they have, and gay men should be more POSITIVELY evaluated the more sexual partners they have. But the authors argue the reverse: “highly active lesbian women would be evaluated positively and highly sexually active gay men negatively, with both being evaluated more negatively than less sexually active gay men and lesbians.” (Note: they are actually asking about ONE sexually active person in their questionnaire, not sexually active men or women in general.) I guess the idea is that gay men are generally more negatively evaluated than lesbians, but highly sexual gay men have one strike against them (gender nonconforming) whereas highly sexual lesbians have two (gender nonconforming and having too much sex).

I would like the authors to clarify how they are deriving their hypotheses. This is very important, as the hypotheses are the crux of the study.

73: Should be “gender-nonconforming”, as in line 71 to be consistent.

81-82: Move this sentence to be the first sentence of the next paragraph.

83-92: The authors argue people who are more masculine will be more supportive of having lots of sexual partners, those who are more feminine less so. But masculine men can be very traditional about sex, at least in the US. Consequently, they might view sex outside of marriage, for example, as bad. (There are likely plenty of studies to examine this issue.) Thus, I’m not convinced that the “masculine” ethos of men is unidimensional in relation to having sex with multiple partners. Perhaps the authors could contextualize their ideas about masculinity more?

It’s also interesting that the authors don’t use the actual sex of the participant as a factor—was there a difference between masculine men and women, etc.?

133-135: The authors note that the dependent variables they are examining have been shown to be sensitive to SDS, but the reference they cite (Marks & Fraley) did not find this to be so when individuals evaluated men and women (as in the current study), only when collaborative groups evaluated them.

134: “have been demonstrated sensitivity to”—fix!

309-313: “the current research was focused on members of one group being evaluated by members of a different group. It would be interesting to examine whether the SDS manifests similarly in non-heterosexual respondents when they are evaluating members of a non-heterosexual population.” But according to line 118, 91% were heterosexual, so 9% were non-heterosexual. Perhaps the authors should use only the heterosexuals in the study? I suspect the nonheterosexuals are less concerned with gender and sexual norms than are the heterosexuals.

Overall, I’m confused by the argumentation the authors use to come up with their predictions. If this could be satisfactorily clarified, I’m fine with the manuscript being published.

Reference cited:

Bell, A. P., Weinberg, M. S., & Hammersmith, S. K. (1982). Sexual preference: Its development in men and women. Bloomington: Indiana University Press.

Horn, S. S. (2007). Adolescents’ acceptance of same-sex peers based on sexual orientation and gender expression. Journal of Youth and Adolescence, 36, 363-371.

Author Response

Reviewer 1

First, we would like to thank Reviewer 1 for their helpful and insightful comments, which were of great help to improving the manuscript.

Comments 1: 52-57: “Similar research on men suggests that those who adhere to traditional gender roles are less satisfied with their romantic relationships than those who adopt unconventional gender roles. Women, however, are more satisfied when their partners adhere to traditional masculine roles [14]. This suggests that although an individual may not be satisfied with adhering to traditional gender roles, their partners prefer that they do.”

I don’t see how the final sentence follows, and I’m also confused about the second sentence.

The first sentence says that masculine/traditional men less satisfied in relationship than unmasculine/feminine/untraditional men; it says nothing about what their partners prefer. The second sentence says that women like more masculine men (or feminine women?). The third sentence says that these ideas suggest that individuals prefer that their partner be traditionally masculine if a man (or traditionally feminine if a woman?—is that found in the research?).  But this seems only true for women, from what was written. Also, it is only masculine/traditional men who are not satisfied with their romantic relationships; an earlier sentence (lines 49-52) says that traditional feminine women are less satisfied in their lives (and says nothing about their relationships). The linkages between ideas are, thus, a bit messy here.

Response 1: We apologize for the inclusion of this section; upon revisiting it, we realize it is not only difficult to follow, but it is also not particularly relevant to the research question and hypotheses. We have replaced it with an expanded discussion on masculine and sexual norms, providing additional clarification as to why we predict a reversed SDS.  (Lines 51-62)

Comments 2: 58-67: The idea here is that, because gay men and lesbians, in being gay and lesbian, go against gender and (for lesbians with many sex partners) sexual norms, the sexual double standard may apply differently to them than to heterosexuals, for whom men having lots of sex is fine, women doing so not fine.

We are told that lesbians receive less verbal harassment than gay men, and are more accepted, so should be less negatively affected by sexual double standard than are gay men.

The authors state that “Gay men and lesbian women are less likely to conform to traditional gender roles [compared to heterosexuals], and thus may be evaluated negatively (compared to heterosexuals) by others” (lines 75-76). But we are not told if John (the gay man) or Joanna (the lesbian) is feminine or masculine, so participants have to rely on stereotype to interpret John and Joanna as gender nonconforming.

Response 2: This is a valid point; Although the ambiguity of whether the target persons were masculine or feminine may have led to some noise in the data, we make the (hopefully shared) assumption that this ambiguity averaged out to zero across the 157 participants. Even so, the notion that people use stereotypes (schemas) to evaluate sexually active others is a core idea—that the SDS is based on stereotypes of sexually active people (e.g., Marks, 2008). Nonetheless, we have expanded the section on gender nonconformity to lend further support to our hypotheses. (Lines 64-70)

Comments 3: Some issues:

(1) The authors’ assumption that sexual orientation and gender nonconformity are perceived the same seems belied in research when these are separated:  “Most surprisingly, for boys, the straight individual who was non-conforming in appearance was rated less acceptable than either the gay individual who conformed to gender norms or was gender non-conforming in choice of activity.” (Horn, 2007, p. 363).

(2) Bell, Weinberg & Hammersmith (1982) found that most gay men and lesbians were gender conforming (did not stand out as gender nonconforming); I think about 23% of their male participants (not randomly selected) were gender nonconforming, but don’t quote me. I think the original Kinsey studies also found that most MSM and WSW were normative gender-wise.

All this means that the authors are working on the idea that their participants have a gender-nonconforming stereotype of gay men and lesbians, but they may not. The authors’ questionnaire pushes this stereotype more for the lesbian than for the gay man, as a business degree seems more “masculine”. This might be discussed as a limitation.

(3) Lesbians violate gender norms (by not being chaste, not being sexually passive AND by being lesbian AND by being masculine?), and by having sex with more people violate sexual roles (women are not supposed to want sex).

Gay men violate gender norms (by being feminine (?) and by being gay), but having sex with more people supports sexual roles (men are supposed to have lots of sex).

From all this I would expect that lesbians should be MORE negatively evaluated the more sexual partners they have, and gay men should be more POSITIVELY evaluated the more sexual partners they have. But the authors argue the reverse: “highly active lesbian women would be evaluated positively and highly sexually active gay men negatively, with both being evaluated more negatively than less sexually active gay men and lesbians.” (Note: they are actually asking about ONE sexually active person in their questionnaire, not sexually active men or women in general.) I guess the idea is that gay men are generally more negatively evaluated than lesbians, but highly sexual gay men have one strike against them (gender nonconforming) whereas highly sexual lesbians have two (gender nonconforming and having too much sex).

I would like the authors to clarify how they are deriving their hypotheses. This is very important, as the hypotheses are the crux of the study.

Response 3: With respect to point 1, non-heterosexuality is socially viewed as a proxy for gender nonconformity (however problematic that may be; Kimmel & Llewellyn, 2012). Physical appearance is something different altogether; there are several conceptualizations of what is considered “gender nonconformity.” As we do not include physical descriptors of the target individuals we hope this is not a large issue.

With respect to point 2, we agree that the business degree held by the target could be non equivalent with respect to a male and female target person; hence, we note this in the limitations section as suggested.  (Lines 336-337)

With respect to point 3, we have expanded the discussion of perceptions of gays and lesbians (as noted above), hopefully clarifying that it is not simply about the number of role violations, but the weight attributed to these violations (e.g., lesbians are eroticized, gay men are perceived as proliferators of AIDS). (Lines 59-61; 85-87)

Finally, we have also changed the hypotheses to refer to a single target person, not men and women generally, as suggested.  (Lines 100-108)

Comments 4: 73: Should be “gender-nonconforming”, as in line 71 to be consistent.

Response 4:  We have made this correction. (Line 76)

Comments 5: 81-82: Move this sentence to be the first sentence of the next paragraph.

Response 5:  We have made this correction. (Lines 89-90)

Comments 6: 83-92: The authors argue people who are more masculine will be more supportive of having lots of sexual partners, those who are more feminine less so. But masculine men can be very traditional about sex, at least in the US. Consequently, they might view sex outside of marriage, for example, as bad. (There are likely plenty of studies to examine this issue.) Thus, I’m not convinced that the “masculine” ethos of men is unidimensional in relation to having sex with multiple partners. Perhaps the authors could contextualize their ideas about masculinity more?

It’s also interesting that the authors don’t use the actual sex of the participant as a factor—was there a difference between masculine men and women, etc.?

Response 6:  Thank you for this comment. First, after reading the reviewer’s comment while referencing the respective selection in our manuscript, we realize we had no grounds on which to state that more feminine individuals may hold negative attitudes about sex. The paragraph was referencing masculine norms relative to femininity; not in contrast to it. We have therefore deleted the last part of the sentence about femininity, since this is not related to our rationale for our hypotheses.  

Second, we argued that masculine individuals would be more supportive of having many sex partners, not that they would necessarily engage in that behavior; Much like people can be sex positive without engaging in sex frequently (or at all). People who are masculine tend to endorse masculine stereotypes (such as valuing sexual prowess); we now convey this in the manuscript. (Lines 51-56)

Finally, although we agree it would be interesting to use participant gender as a factor in the regressions, we did not have nearly the amount of statistical power required to do so--Testing the SDS as moderated by gender identity and participant gender would require the testing of four-way interactions. (We do note this in the limitations section as part of our response to Reviewer 2.) (Lines 331-334)

Comments 7: 133-135: The authors note that the dependent variables they are examining have been shown to be sensitive to SDS, but the reference they cite (Marks & Fraley) did not find this to be so when individuals evaluated men and women (as in the current study), only when collaborative groups evaluated them.

Response 7:  Apologies; our wording choice made this confusing. We meant to convey that the items are sensitive to variables central to the SDS (gender and sex partners)—not the SDS itself. We have edited the language here to make this clearer. (Also, as a point of clarification, the work we referenced was the 2005 Marks and Fraley paper, not the 2007 research that examined differences in SDS emergence between individuals and collaborative groups). (Lines 147-148)

Comments 8: 134: “have been demonstrated sensitivity to”—fix!

Response 8: We have made this correction. (Line 147)

Comments 9: 309-313: “the current research was focused on members of one group being evaluated by members of a different group. It would be interesting to examine whether the SDS manifests similarly in non-heterosexual respondents when they are evaluating members of a non-heterosexual population.” But according to line 118, 91% were heterosexual, so 9% were non-heterosexual. Perhaps the authors should use only the heterosexuals in the study? I suspect the nonheterosexuals are less concerned with gender and sexual norms than are the heterosexuals.

Response 9:  We have re-run the analyses on heterosexual individuals only, and the results were almost exactly the same with respect to which effects are significant. We now note this in the manuscript, and while we did not write up and insert the results of this analysis in the service of reducing redundancy and saving space, we note that these analyses are available by contacting the first author. (Lines 349-355)

Reviewer 2 Report

Comments and Suggestions for Authors

Thank you very much for giving me the opportunity to review this manuscript titled "The sexual double standard toward non-heterosexual populations: evaluations of sexually active gay men and lesbian women". I would like to take this opportunity to congratulate the authors for their efforts.  

I would also like to point out that the manuscript has clear shortcomings that compel me to recommend rejection of publication. I would like to point out the limitations of this manuscript. 

Introduction 

The sexual double standard that the authors deal with is really about sexual attitudes. For that reason, in the introduction, they should have included this topic. The introduction needs to be improved. 

Hypotheses need to be better described. 

Materials and Methods 

The authors write: "G*Power software suggested tha an approximate minimum of 132 participants would be necessary...". This is not correct, G*Power does not suggest a minimum, but points to the sample size. For this reason, the authors should indicate why they exceeded the sample size. Authors should also indicate the parameters used for the calculation of the sample in G*Power, so that the calculation can be replicated. 

The authors do not describe the questionnaire for the collection of socio-demographic data. The only reference to this questionnaire is in the Procedure section, but the questionnaire is not described. 

The PAQ questionnaire was designed in 1979. The authors themselves include this information in the limitations. Many social changes have taken place since then. The questionnaire should have been updated. 

The double sexual standard questionnaire used is an adaptation of the original, but the validity of the adapted form has not been confirmed. This is a serious methodological error. The new form should be adapted. In addition, the reference in APA format is included.  

Results 

Results should be presented on the basis of the hypotheses, not on the basis of the information in the questionnaire. 

The results should be modified as indicated. 

Discussion 

The authors write:  "Our findings reflect the trend of negative evaluations for most highly sexually active individuals". There is insufficient evidence for this claim. The problems in methodology, noted above, preclude this assertion. 

The authors write: "In conjuncture with the daily stress of being non-heterosexual, this could be associated with developing more serious mental disorders and suicide" . There is insufficient evidence for this claim. Moreover, it is a dangerous assertion. Conclusions should be drawn with more caution. 

The authors point out in the limitations that they do not have sufficient statistical power for the study, therefore, the results should be treated with more caution.  

In the limitations, the authors do not point out the differences in the sample (67% women), which may also affect the results. 

Author Response

Reviewer 2

First, we would like to thank Reviewer 2 for their helpful and insightful comments, which were of great help to improving the manuscript.

Comments 1: The sexual double standard that the authors deal with is really about sexual attitudes. For that reason, in the introduction, they should have included this topic. The introduction needs to be improved. 

Response 1: Our apologies; in discussing this issue, we (the authors) were unable to say with any certainty what the issue with respect to a lack of discussion of attitudes. The way we define the SDS in the current research is with respect to people’s evaluations of sexually active others. We, as social psychologists, use the term “attitude” to describe relatively enduring evaluations of a target, or rather, the attitude object. The attitude objects in this case are sexually active gay men and lesbian women. When we state that attitudes are evaluations, we mean that they involve a preference for or against the attitude object, as commonly expressed in terms such as “prefer,” “like,” etc. This issue aside, we hope that the revisions to the introduction that were made in response to Reviewer 1’s concerns have sufficiently improved the introduction. (Lines 43-90)

Comments 2: Hypotheses need to be better described. 

Response 2: We hope the revisions made in response to Reviewer 1 are a sufficient response to this concern. (Lines 100-108)

Comments 3: The authors write: "G*Power software suggested that an approximate minimum of 132 participants would be necessary...". This is not correct, G*Power does not suggest a minimum, but points to the sample size. For this reason, the authors should indicate why they exceeded the sample size. Authors should also indicate the parameters used for the calculation of the sample in G*Power, so that the calculation can be replicated. 

Response 3: We have elaborated on the power analysis, including all of the parameters used to generate the sample size suggestion. The final sample is slightly higher than this because we anticipated attrition, and we thus overrecruited participants to avoid having to assess attrition and re-open the study multiple times until 132 participants were attained. We now elaborate on the procedure in the methods. We also conducted a post-hoc power analysis to assuage concern that the final sample of 157 was not excessively over powered. Results showed power with 157 participants was .93, three percent over our initial goal of .90.  (Lines 120-127)

Comments 4: The authors do not describe the questionnaire for the collection of socio-demographic data. The only reference to this questionnaire is in the Procedure section, but the questionnaire is not described.

Response 4: The demographics items were not a separate measure per se; they were basic demographic items surveyed in most psychological research (age, gender, ethnicity, sexual orientation, and relationship status). We feel the description of the sample in the “participants” section is self-explanatory as to what was asked, thus we did not list the items as to avoid redundancy. However, if the editor would like, we will provide the demographic items in an appendix.

Comments 5: The PAQ questionnaire was designed in 1979. The authors themselves include this information in the limitations. Many social changes have taken place since then. The questionnaire should have been updated. 

Response 5: We indeed acknowledge the age of the PAQ. Given this, and given that the research has been completed, we felt the optimal route to assuage concern about the age of the PAQ was to review the literature for evidence that the PAQ is still a valid contemporary assessment of gender self-concept. We have updated the discussion to include citations to attest to this fact. (Lines 325-329)

Comments 6: The double sexual standard questionnaire used is an adaptation of the original, but the validity of the adapted form has not been confirmed. This is a serious methodological error. The new form should be adapted. In addition, the reference in APA format is included.  

Response 6: We now report internal consistencies for each of the 5 scale dependent variables. The APA reference has also been corrected. (Lines 144-146)

Comments 7: Results should be presented on the basis of the hypotheses, not on the basis of the information in the questionnaire. The results should be modified as indicated. 

Response 7: We understand how the current format of the results may be nonintuitive with respect to testing the hypotheses, and that it is the status quo to report analyses by hypothesis. After much consideration of how we could clearly present results by hypothesis, we have decided to keep the results as they are (by area) rather than by hypothesis. Our rationale is that the hypotheses each focus on one regression coefficient, which would result in either repeating each of the 5 regressions, or only presenting partial results for each regression by hypothesis, which would, in our opinion, be more difficult to follow. Hence, the options were to either induce redundancy or to present partial regressions for each hypothesis, neither of which flow as well and are as intuitive as they are currently presented. Nonetheless, to assuage this legitimate concern and make it clearer as to whether the hypotheses were supported, we now present summaries by hypothesis in a new heading in the results section. (Lines 249-260)

Comments 8: The authors write:  "Our findings reflect the trend of negative evaluations for most highly sexually active individuals". There is insufficient evidence for this claim. The problems in methodology, noted above, preclude this assertion. 

Response 8: We are unsure whether the Reviewer means that there is insufficient evidence in the literature to support this claim, or that there is insufficient evidence in our data to support this claim. With respect to the former, we now include several citations corroborating that highly sexually active individuals are, on average, evaluated negatively. With respect to the latter, for each of the five regressions, there was a negative regression coefficient for the “number of partners” variable, two of which were highly significant and the remaining three were marginally significant. We believe this consistency is justifiable evidence that corroborates our statement about highly sexually active individuals being evaluated negatively in our sample.

Comments 9: The authors write: "In conjuncture with the daily stress of being non-heterosexual, this could be associated with developing more serious mental disorders and suicide" . There is insufficient evidence for this claim. Moreover, it is a dangerous assertion. Conclusions should be drawn with more caution. 

Response 9: We agree that the claim made here is too strong and there is little evidence to imply this. We therefore have removed this claim.  

Comments 10: The authors point out in the limitations that they do not have sufficient statistical power for the study, therefore, the results should be treated with more caution.  

Response 10: We stated we did not have statistical power to test 4-way interactions, not that we didn’t have power for the study. We do acknowledge there were few significant effects, however, so we have noted that the results should indeed be interpreted with caution. (Lines 330-336)

Comments 11: In the limitations, the authors do not point out the differences in the sample (67% women), which may also affect the results. 

Response 11: We have now added this caveat to the limitations section following other caveats about the analyses/results. (Lines 334-336)

Round 2

Reviewer 1 Report

Comments and Suggestions for Authors

The authors have dealt effectively with my concerns.

Reviewer 2 Report

Comments and Suggestions for Authors

Thank you to the authors for their responses. 

Despite slight discrepancies with some responses, I think the manuscript has improved considerably. Congratulations for the effort made.